# MicroRNAs in Small Extracellular Vesicles from Amniotic Fluid and Maternal Plasma Associated with Fetal Palate Development in Mice

**DOI:** 10.3390/ijms242417173

**Published:** 2023-12-06

**Authors:** Xige Zhao, Xia Peng, Zhiwei Wang, Xiaoyu Zheng, Xiaotong Wang, Yijia Wang, Jing Chen, Dong Yuan, Ying Liu, Juan Du

**Affiliations:** 1Laboratory of Orofacial Development, Laboratory of Molecular Signaling and Stem Cells Therapy, Molecular Laboratory for Gene Therapy and Tooth Regeneration, Beijing Key Laboratory of Tooth Regeneration and Function Reconstruction, Capital Medical University School of Stomatology, Tiantan Xili No. 4, Beijing 100050, China; 18842663996@163.com (X.Z.); 18834183343@163.com (X.P.); w18835570649@163.com (Z.W.); 18519220100@163.com (X.Z.); wangxiaotongkx98@163.com (X.W.); yijiawang1994@163.com (Y.W.); chenjingecho629@163.com (J.C.); ying_liu202311@163.com (Y.L.); 2Department of Geriatric Dentistry, Capital Medical University School of Stomatology, Tiantan Xili No. 4, Beijing 100050, China; yuand_y@126.com

**Keywords:** small extracellular vesicles, palate development, amniotic fluid, plasma, microRNA

## Abstract

Cleft palate (CP) is a common congenital birth defect. Cellular and morphological processes change dynamically during palatogenesis, and any disturbance in this process could result in CP. However, the molecular mechanisms steering this fundamental phase remain unclear. One study suggesting a role for miRNAs in palate development via maternal small extracellular vesicles (SEVs) drew our attention to their potential involvement in palatogenesis. In this study, we used an in vitro model to determine how SEVs derived from amniotic fluid (ASVs) and maternal plasma (MSVs) influence the biological behaviors of mouse embryonic palatal mesenchyme (MEPM) cells and medial edge epithelial (MEE) cells; we also compared time-dependent differential expression (DE) miRNAs in ASVs and MSVs with the DE mRNAs in palate tissue from E13.5 to E15.5 to study the dynamic co-regulation of miRNAs and mRNAs during palatogenesis in vivo. Our results demonstrate that some pivotal biological activities, such as MEPM proliferation, migration, osteogenesis, and MEE apoptosis, might be directed, in part, by stage-specific MSVs and ASVs. We further identified interconnected networks and key miRNAs such as miR-744-5p, miR-323-5p, and miR-3102-5p, offering a roadmap for mechanistic investigations and the identification of early CP biomarkers.

## 1. Introduction

Cleft palate (CP) is a prevalent congenital birth defect [1]. Both genetic and environmental factors that impede palatogenesis can result in CP [2]. This defect can significantly impact quality of life, even following surgical interventions, and places a considerable strain on families and society. The effectiveness of ultrasound-mediated diagnosis can be compromised by various factors such as maternal obesity, fetal position, and operator proficiency. Often, the palate remains inadequately visualized due to acoustic shadowing [3]. As a result, it is important to deepen our understanding of the biological processes underpinning palatogenesis. This would aid the advancement of potential prenatal diagnostic biomarkers and interceptive therapies.

Recent studies indicate that maternal conditions can influence fetal development via microRNAs (miRNAs). There are a set of short noncoding RNAs that post-transcriptionally modify gene expression [4,5]. An increasing body of evidence points to the role of miRNAs in processes such as proliferation, apoptosis, and cell migration during palatogenesis [6,7,8]. Previously, we identified miRNA expression profiles in miniature pigs during palatogenesis. Notably, a gene ontology (GO) term analysis indicated that small extracellular vesicles (SEVs) ranked within the top three significant GO terms at all stages [9]. This infers a possible role for miRNAs in palate development via SEVs and has drawn our attention to their potential involvement in palatogenesis.

Containing a range of RNAs (particularly miRNAs), SEVs are a kind of cell-to-cell communication medium. These vesicles can be detected in numerous body fluids, notably maternal plasma (MP) and amniotic fluid (AF). Both SEVs derived from AF (ASVs) and MP (MSVs) are in direct contact with the embryo and undergo changes in composition and instructional capacities throughout their development [10,11,12]. Non-invasive assays of ASVs and MSVs have shown promise for development as they are potentially predictive, sensitive, and robust [13]. Furthermore, compared to free-floating miRNAs in MP and AF, vesicle-encapsulated miRNAs are shielded from rapid degradation in the bloodstream. This facilitates their transfer to target cells [14]. These miRNAs not only provide insights into the functional state of their source cells but also have the potential to alter the phenotype of the recipient cells [15]. Therefore, profiling miRNAs found within ASVs and MSVs could help devise prognostic biomarkers and therapeutic strategies for various birth defects.

Some studies have attempted to identify MSV-associated risk factors for cleft lip with or without palate (CL/P) through the direct sequencing of genomes from patients with CL/P [11,16,17]. While a few miRNAs in MSVs have demonstrated diagnostic potential [11,16], the molecular mechanisms remain unclear due to the absence of appropriate methodologies to analyze normal embryos during craniofacial developmental stages. While studies on miRNAs in ASVs have helped to clarify specific fetal changes and diseases, such as severe congenital diaphragmatic hernia [18] and fetal alcohol syndrome [19], their roles in craniofacial development or orofacial cleft remain largely unexplored.

Mice are the predominant animal model used to investigate palate development [20]. Palate development in mice commences with the formation of the palatal shelves on embryonic day 11.5 (E11.5). From E12.5 to E13.5, the shelves ascend vertically on both sides of the tongue. They subsequently elevate to a horizontal position between E13.5 and E14.5, progressing toward the midline to fuse which initiates at E14.5 and completes by E16.5. This sequence bears significant resemblance to palate development in humans [20]. Within this progression, the stages from E13.5 to E15.5 are primarily examined to understand the etiology of CP [21,22,23]. However, there have been no reports about miRNA profiles in the ASVs/MSVs of mice during palate development, which may influence the palatogenesis through the biological behaviors of palatal cells.

In this study, we used an in vitro model to determine how ASVs/MSVs influence the biological behaviors of mouse embryonic palatal mesenchyme (MEPM) cells and medial edge epithelial (MEE) cells, and then investigated the time-dependent differential expression (DE) of miRNAs in the ASVs and MSVs of mice at E13.5/E14.5/E15.5 via pairwise comparison in vivo, and compared these miRNAs with the time-dependent DE mRNAs in the palates at E13.5/E14.5/E15.5 to study the dynamic co-regulation between the miRNAs in the ASVs/MSVs and the mRNAs in the palates during palatogenesis. To the best of our knowledge, this is the first study to examine the roles of SEVs in standard palatogenesis. It further suggests prospective prenatal diagnostic biomarkers and intervention strategies for susceptible populations, facilitating the investigation of fluid–tissue interactions during palate development.

## 2. Results

### 2.1. Morphology and Histology of Palate at E13.5/E14.5/E15.5

To observe whether the development time of collected embryos was accurate, two embryos from each pregnant mouse were selected for morphological observation. The developmental stages of the secondary palate in the mice are shown in Figure 1A. Sequential histological sections obtained from E13.5 to E15.5 were stained with hematoxylin and eosin (HE). Each embryo exhibited typical stage-matched characteristics. At E13.5, the palatal shelves displayed defined shapes and then transitioned to a horizontal orientation. By E14.5, they had converged at the midpalate epithelial suture (MES), initiating the fusion process. By E15.5, the fusion was complete, and a pronounced osteogenic center had formed.

### 2.2. Isolation and Identification of MSVs and ASVs

SEVs were isolated from the MP and AF of pregnant mice at E13.5, E14.5, and E15.5 via ultracentrifugation (Figure 1B). The vesicles displayed the characteristic cup-shaped morphology (Figure 1C; Appendix A), with particle diameters between 70 and 160 nm (Figure 1D; Appendix A). The SEV markers TSG101, CD63, and CD9 were also detected (Figure 1E). These findings indicate that the isolation methods used for MSVs and ASVs are effective, producing relatively pure SEVs.

### 2.3. Different Effects of MSVs and ASVs on the Biological Behaviors of MEPM Cells and MEE Cells

MEE cells and MEPM cells are the major cells in palate shelves [24]. In the early phase of palate development, the growth and motility of the palatal shelves are determined mainly by the proliferation and migration of MEPM cells [25], while late osteogenesis depends on the osteogenic differentiation of MEPM cells [26]. Therefore, we first investigated the roles of MSVs and ASVs in MEPM cell proliferation, migration, apoptosis, and osteogenesis. We first labeled the SEVs with green fluorescence and co-cultured them with MEPM cells for 24 h. Both ASVs and MSVs were taken up by MEPM cells (Figure 2A). Cell counting kit 8 (CCK8) showed that the E13.5 MSVs had the strongest effect on MEPM cell proliferation, followed by the E14.5 MSVs, E15.5 MSVs, E13.5 ASVs, E14.5 ASVs, and E15.5 ASVs (Figure 2B). A scratch assay showed that the E13.5 MSVs also enhanced MEPM migration the most, followed by the E14.5 MSVs, E15.5 MSVs, and E13.5 ASVs (Appendix A; Figure 2C). Flow cytometry further showed that the E13.5 MSVs inhibited MEPM cell apoptosis, while the other groups of SEVs did not have obvious effects on MEPM apoptosis (Figure 2D,E). Next, alizarin red staining showed that the E15.5 ASVs, E14.5 ASVs, and E15.5 MSVs induced osteogenesis in the MEPM cells (Figure 3A,B).

The decomposition of the MES is mainly related to MEE apoptosis in the middle phase of palate development [27]. Thus, we further explored the effects of MSVs and ASVs on MEE cell apoptosis. We also labeled the SEVs with green fluorescence and co-cultured them with MEE cells for 24 h. Both the ASVs and MSVs were taken up by MEE cells (Figure 3C). We then found that the E14.5 ASVs most notably augmented the apoptosis of MEE cells, followed by E15.5 ASVs (Figure 3D,E). E14.5 MSVs and E15.5 MSVs did not have significant effects on MEE apoptosis, while E13.5 ASVs and E13.5 MSVs inhibited MEE apoptosis to an extent.

In summary, E13.5 MSVs most significantly enhanced the proliferation and migration of MEPM cells, while E14.5 ASVs most notably augmented the apoptosis of MEE cells, and E15.5 ASVs induced osteogenesis in MEPM cells.

### 2.4. Analysis of miRNA Sequences in ASVs and MSVs from Different Stages

To find the main components of SEVs regulating palatal cell biological behavior, we analyzed the miRNA sequences in the ASVs and MSVs from different stages. We identified 1054 miRNAs in ASVs and 653 miRNAs in MSVs. Overall, 772 (73.2%) miRNAs were co-expressed across all three stages in ASVs and 438 (67%) miRNAs were co-expressed across all three stages in MSVs (Figure 4A). Because embryonic development is a specific, spatiotemporal process, and palatogenesis in mice differs across the three stages, next, we focused on identifying significant DE miRNAs at E13.5/E14.5/E15.5 via pairwise comparison. Out of 308 DE miRNAs, 48 were upregulated, and 177 were downregulated between E13.5 and E14.5 in ASVs whereas 5 and 101 were in MSVs, respectively. Out of 274 DE miRNAs, 110 were upregulated and 126 were downregulated between E14.5 and E15.5 in ASVs, whereas 26 and 19 were in MSVs, respectively. Out of 436 DE miRNAs, 114 were upregulated and 212 were downregulated between E13.5 and E15.5 in ASVs, whereas 7 and 156 were in MSVs, respectively (Figure 4B). The significantly DE miRNAs between E13.5 ASVs and E14.5 ASVs included miR-9-5p, miR-122-5p, miR-744-5p, and miR-3473b (Appendix A). And miR-9-5p, miR-1298-5p, and miR-3102-5p were the significantly DE miRNAs between E14.5 ASVs and E15.5 ASVs (Appendix A). Interestingly, miR-9-5p was also the most significantly DE miRNA between E13.5 ASVs and E15.5 ASVs (Appendix A). On the other hand, the significantly DE miRNAs between E13.5 MSVs and E14.5 MSVs included miR-294-3p, miR-138-5p, and miR-323-5p (Appendix A), while miR-181c-5p, miR-483-3p, and miR-127-3p were among the significantly DE miRNAs between E14.5 MSVs and E15.5 MSVs (Appendix A). And miR-541-5p, miR-431-3p, and miR-3072-3p were the significantly DE miRNAs between E13.5 MSVs and E15.5 MSVs (Appendix A). These DE miRNAs are related to functions such as the regulation of multicellular organism development and system development (Appendix A).

Heatmaps from the three stages showed that DE miRNAs with the most significant differences. For the ASVs, the miRNA expression levels varied the most between E13.5 and E15.5, followed by E14.5 and E15.5, and finally between E13.5 and E14.5 (Figure 4C). Conversely, in the MSVs, they differed most significantly between E13.5 and E14.5, followed by E13.5 and E15.5, with only a minor difference between E14.5 and E15.5 (Figure 4D). These findings are consistent with the Venn diagrams (Figure 4B).

### 2.5. Analysis of DE miRNAs in ASVs and MSVs

A volcano map of the comparison of ASVs and MSVs at the same stage showed that the miRNA expression levels varied the least between the E13.5 ASVs and MSVs, followed by E14.5 and E15.5 (Figure 5A–C). The significantly DE miRNAs at E13.5 included miR-544-5p, miR-449c-5p, miR-744-5p, and miR-138-5p (Figure 5A), while the significantly DE miRNAs at E14.5 included miR-743a-3p, miR-127-3p, miR-16-5p, and miR-880-3p (Figure 5B), and miR-344-3p, miR-138-5p, miR-298-5p, and miR-3072-3p were the significantly DE miRNAs at E15.5 (Figure 5C). The dominant functions of these DE miRNAs are the positive regulation of the biological process and cellular process at E13.5, and the positive regulation of the metabolic processes at E14.5 and E15.5 (Figure 5D–F).

### 2.6. Regulatory Networks of miRNAs in ASVs and mRNA in Palate Tissue

As the DE miRNAs of different stages in ASVs or MSVs may be involved in the regulation of multiple organ and system developments, to better understand the phenotypes observed and explore the dynamic effects of DE miRNAs in ASVs and MSVs on palate development in vivo, we introduced E13.5-E15.5 fetal palate tissue transcriptome sequencing data. We integrated time-dependent DE miRNAs in ASVs or MSVs with time-dependent DE mRNAs (Top 100) in fetal palates via pairwise comparison to construct regulatory networks (see Methods for details). While forming these networks, we focused on miRNA–gene pairs that exhibited negative correlations, embodying the repressive nature of miRNAs and mirroring genuine miRNA regulation within the cellular systems [28,29].

In line with Yu et al. [30], we identified three hub miRNAs (miR-3473b, miR3473e, and miR-744-5p) between E13.5 ASVs and E14.5 ASVs (Figure 6A). The main target genes of miR-3473 are sphingomyelin phosphodiesterase 3 (*Smpd3*), angiotensinogen (*Agt*), and the phosphate-regulating gene with homologies to endo-peptidases on the X chromosome (*Phex*). The target genes of miR-744-5p include refilin A (*Rflna*), forkhead box C2 (*Foxc2*), and matrilin 4 (*Matn4*). The corresponding functions identified in the GO assay were mainly linked to the regulation of the extracellular matrix organization (*Smpd3*, *Agt*, and *Foxc2*) and ossification (*Phex*, *Foxc2*, *Rflna*, and *Smpd3*) (Figure 6B). Genes associated with pathways including protein digestion and absorption were significantly enriched in the Kyoto Encyclopedia of Genes and Genomes (KEGG) assay (Appendix A).

The hub miRNA, miR-3102-5p, was a DE miRNA between E14.5 ASVs and E15.5 ASVs (Figure 6C). The target genes of miR-3102-5p include P2X purinoceptor 4 (*P2rx4*) and decorin (*Dcn*). The corresponding GO assay functions were mainly linked to the regulation of endothelial cell migration (*P2rx4* and *Dcn*), the cellular response to ATP (*P2rx4*), and the positive regulation of mitochondrial depolarization (*Dcn*). In addition, retinal metabolism (retinol-binding protein 4 (*Rbp4*) targeted by miR-880-3p, miR-292-3p, and miR-16-5p), which is critical to palate development, was also enriched in the top five significant biological process terms in the GO analysis (Figure 6D). Genes associated with the TGF-beta signal pathway were significantly enriched in the KEGG assay (Appendix A).

Another hub miRNA (miR-673-5p) was a DE miRNA between E13.5 ASVs and E15.5 ASVs (Figure 6E). The target genes of miR-673-5p include lysyl oxidase (*Lox*). The corresponding GO assay functions were mainly linked to extracellular matrix organization (*Lox*) and the transforming growth factor beta receptor signal pathway (*Lox*). In addition, cilium movement (dynein axonemal heavy chain 10 (*Dnah10*) targeted by miR-7071-5p), which is critical to palate development, was also enriched in the top 10 significant biological process terms in the GO analysis (Figure 6F). Genes associated with pathways including motor proteins were significantly enriched in the KEGG assay (Appendix A).

### 2.7. Regulatory Networks of miRNAs in MSVs and mRNA in Palate Tissue

Eleven hub miRNAs (especially miR-138-5p, miR-323-5p, miR-370-3p, and miR-3072-3p) were identified between E13.5 MSVs and E14.5 MSVs (Figure 7A). The target genes of miR-138-5p include *Agt* and sphingomyelin synthase 2 (*Sgms2*). The target genes of miR-323-5p include collagen XXII (*Col22a1*) and dynein axonemal light intermediate chain 1 (*Dnali1*). The target genes of miR-370-3p include *Lox* and *Lox14*. The target gene of miR-3072-3p is *Dnah10*. The corresponding GO assay functions were mainly linked to cilium movement/microtubule-based movement (*Dnah10* and *Dnali1*) and extracellular matrix organization (*Agt*, *Col22a1*, *Lox*, *Loxl4*, and *Smpd3*) (Figure 7B). Genes associated with pathways including ECM-receptor interaction were significantly enriched in the KEGG assay (Appendix A).

There was no hub miRNA between E14.5 MSVs and E15.5 MSVs. Only three miRNA-mRNA pairs were involved (Figure 7C). The corresponding GO assay functions were mainly linked to myotube differentiation, which is involved in skeletal muscle regeneration (cluster of differentiation 9 (*Cd9*)) and the regulation of platelet aggregation and activation (*Cd9*, desmoglein 3 (*Dsg3*)) (Figure 7D). Genes associated with the pathway were in the hematopoietic cell lineage (Appendix A).

Seven hub miRNAs (especially miR-138-5p, miR-298-5p, miR-431-3p, and miR-3072-3p) were identified between E13.5 MSVs and E15.5 MSVs. The target gene of miR-138-5p is *Sgms2*. The target genes of miR-298-5p include osteomodulin (*Omd)* and *Sgms2*. The target gene of miR-431-3p is *Dnah10* (Figure 7E). The target genes of miR-3072-3p include *Dnah10* and *Hydin*. The corresponding GO assay functions were mainly linked to bone mineralization (*Lox*, *Omd*, and *Sgms2*) and cilium movement (*Dnah10* and *Hydin*) (Figure 7F). Genes associated with pathways including motor proteins were significantly enriched in the KEGG assay (Appendix A).

### 2.8. Quantitative Analysis of miRNA Expression

The validation of expression patterns from the quantity of captured sequence data was tested via quantitative real-time polymerase chain reaction (qRT-PCR). We selected six DE miRNAs, which also played important roles in the regulatory networks. Their expression patterns agreed with the patterns determined using high-throughput sequencing. The findings verified that the miRNA sequencing data were reliable and could be further analyzed (Figure 8).

## 3. Discussion

Recently, SEVs, including their content, were detected to communicate with cells by working as shuttles of molecules to influence target cells with biomarkers (miRNAs, proteins, and lipids), which can be released in the extracellular environment, both in pathological and physiological conditions. Exploring specific SEV content, which plays a key role in the pathophysiological conditions of patients or in the development of exosomes as shuttles for local drug release and their content for targeted therapy, is promising in the clinic [31]. In a study conducted by Jia et al., an analysis of the MSVs from pregnant women carrying either normal fetuses or fetuses with CLP revealed the identification of let-7 clusters as prenatal biomarkers for CLP [16], which shows the importance of the SEVs of pregnant women in detecting an embryo’s CLP. 

Despite the significant strides in discerning the intricate cellular and morphological processes during palatogenesis, our knowledge of the molecular mechanisms instructing this fundamental phase remains limited. In this research, we delve into the characteristics of MSVs and ASVs during palate development in mice and present a novel perspective on their potential roles in developmental events. The development of the fetal palate is meticulously regulated and precisely coordinated. Our morphological analysis underscores that by E13.5, the palatal shelves growing adjacent to the tongue begin reorienting horizontally. By E14.5, they converge at the midline. By E15.5, the palatal suture vanishes, and an osteogenic center emerges. These morphological shifts are in line with prior studies that have highlighted various biological patterns in palatal cells at different development stages. For instance, E13.5 MEPM cells have been found to demonstrate heightened proliferation, aligning with the swift growth of early palate shelves [22,32]. E14 MEE cells have been found to exhibit pronounced epithelial–mesenchymal transition (EMT) and apoptosis, contributing to MES dissolution [33]. By E15.5, MEPM cells show amplified expressions of markers of bone development, hinting at the palate’s osteogenic differentiation potential [22]. Building on these insights, our in vitro study infers that some of these pivotal biological activities might be directed, in part, by stage-specific MSVs and ASVs. Furthermore, our observations hint that MSVs have a pronounced regulatory impact on MEPM cells in the early stage (E13.5), while ASVs become influential on both MEPM cells and MEE cells during the middle and latter stages (E14.5 and E15.5).

We then examined miRNAs across various stages of MSVs and ASVs. The miRNA shifts in ASVs were significant throughout the three stages, whereas differences in the miRNAs within MSVs became more limited after E14.5. In line with this, the volcano map also showed that there were more consistent miRNAs in MSVs and ASVs at E13.5, but they decreased significantly after E14.5. This distinction might arise from the differential origins of the two SEV types. ASVs are primarily sourced from MP, amnion, and placental tissues, transitioning to fetal urine and fetal lung fluid as the organs mature (although the specific origin of murine AF at diverse gestational stages remains undefined, studies indicate that processes such as glomerular filtration and tubular maturation initiate at E14.5 in the kidneys of mice) [34]. By contrast, the bulk of MSVs emanate from maternal cells, with only a fraction deriving from the placenta [35]. Hence, ASVs exhibit greater variations in the composition and regulatory potential than MSVs as major organs progress [34], potentially explaining the limited influence of MSVs on palatal cell patterns in the middle to late stages.

To better understand the phenotype observed and further explore the possible interrelated mechanisms of miRNAs in ASVs and MSVs on palate development in vivo, we compared and integrated DE miRNA in ASVs/MSVs and DE mRNAs in the palate. E13.5–E14.5 was the fastest growth period of palate development in mice and showed the maximum morphological change [21]. Consistent with the phenotype observed in vitro, at this stage, the miRNAs in the ASVs were mainly involved in functions related to extracellular matrix organization, which could regulate cell proliferation, migration, and adhesion to meet the rapid growth of functional requirements [36]. The hub miRNAs in the ASVs during this stage were miR-3473b, miR-3473e, and miR-744-5p. Notably, *Foxc2*, the target gene of miR-744-5p in the network, has been reported to be associated with CP. *Foxc2* is a member of the fork head box family of transcription factors and has been shown to be functional during the development of cranial mesenchymal tissues [37]. The loss of function in *Foxc2* results in craniofacial anomalies, notably CP, but the mechanism is unclear [38]. In the present study, a decline in *Foxc2* expression was observed as the embryo matured, with potential influences from increased miR-744-5p in ASVs. This suggests that the regulation of *Foxc2* expression by miR-744-5p may play an important role in maintaining normal palatal development, and the change in miR-744 expression in ASVs could serve as a biomarker for the early diagnosis of palate dysplasia.

For MSVs between E13.5 and E14.5, miRNAs were also involved in the regulation of extracellular matrix organization. Additionally, they were mainly involved in the regulation of cilium movement and microtubule-based movement, which might explain their powerful role in regulating MEPM migration. Moreover, the primary cilium is a pivotal sensory organelle acting as a major signaling hub for a number of signaling pathways that are essential for craniofacial development, such as Hedgehog [39] and WNT signaling [40]. Cilium dysfunction disrupts multiple signaling pathways and their interactions, resulting in widespread phenotypic defects, collectively called ciliopathy [41]. Particularly, in some studies, SEVs were found to be attached to the surface of the primary cilium [42,43]. An idea for the phenomenon was that the primary cilium is the receptor that receives the SEVs, and accumulating evidence suggested that the primary cilium also functioned as the “emitter” of SEVs [44]. Physiological and pathological impacts were also elucidated for the release of SEVs from primary cilia [45]. These relationships suggest the special role of SEVs in ciliopathy. However, the roles of MSVs in the primary cilium during palate development remain to be clarified. miR-323-5p, involved in both the regulation of cilium movement and extracellular matrix organization, represents the functional characteristics of MSVs at this stage, so it is expected to serve as an entry point for future mechanism studies.

From E14.5 to E15.5, the main event was the fusion process of the palate shelves. During this stage, the MEE cells died, and the dead cells were phagocytosed by macrophages [46,47]. Consistent with this, the miRNA in the ASVs during this stage were involved in the regulation of mitochondrial depolarization, macrophage migration, and many terms related to endothelial cell migration and chemotaxis. Mitochondrial depolarization was important to induce apoptosis and active immune cells in many fetal diseases [48,49], which suggested its potential contribution to MES degradation during this stage. The endothelial cell migration and chemotaxis might play important roles in transporting macrophages to dead phagocytosed MEE cells. In addition, a previous study has confirmed that vascular endothelial growth factor A (VEGFA) plays an important role in palate osteogenesis [50]. This suggests that the ASVs involved in endothelial cell migration and chemotaxis might also regulate the initial osteogenesis during this stage. The hub miRNA, miR-3102-5p, which is involved in both endothelial cell migration and mitochondrial depolarization, represents the most significant functional characteristics of MSVs at this stage, so it may serve as a roadmap for mechanistic investigations in further studies. Moreover, miR-16-5p, miR-292a-3p, and miR-880-3p, which target *Rbp4* and then participate in regulating epithelial structure maintenance and retinal metabolic processes, aroused our concern. It has been shown that retinoic acid (RA) plays important roles during palate development, but excess RA causes CP in the fetuses [51]. Dong et al. suggested that *Rbp4* was involved in CP induced by RA, and it could be suppressed by RA to cause growth inhibition in the embryonic palate [52]. However, the effect of *Rbp4* on epithelial maintenance and its role in palate development await further elucidation. 

For MSVs between E14.5 and E15.5, only three miRNA-mRNA pairs were involved because the changes in the miRNA in the MSVs were very little at this stage. The corresponding GO assay functions were the most significantly linked to myotube differentiation involved in skeletal muscle regeneration, which might be regulated by *Cd9* targeted by miR-127-3p. Rot et al. suggested that the proper fusion of the palatal shelves depends on both mechanical cues and paracrine contributions from the skeletal muscle [53], and disordered palatogenesis has been reported in the absence or damage of skeletal muscle [53,54]. This suggests that MSVs may indirectly affect palate development by regulating skeletal muscle development during the middle and late stages of palate development.

Our previous in vivo study demonstrated that the microenvironment might guide the functional transition of palate tissue from proliferation to osteogenesis [22]. In the present study, the most obvious functional change from E13.5 SEVs to E15.5 SEVs was also confirmed to the transition from promoting MEPM cell proliferation to promoting MEPM cell differentiation. Consistent with these phenotypes, the miRNAs (especially miR-673-5p) in the ASVs during this stage were involved in many items related to the regulation of the TGF-beta signaling pathway, which were essential for MEPM differentiation [55]. Concerning the MSVs, the miRNAs (especially miR-298-5p) were directly involved in regulating bone mineralization during this stage. The enriched networks predominantly intersected with the motor protein pathways in both the ASVs and MSVs, which were related to intracellular component transport and movement, suggesting an active metabolism during this stage. Moreover, we noticed that the pathway in histidine metabolism was specifically enriched in AVSs. Emerging evidence has demonstrated the relationship between maternal histidine metabolism and fetal development [56,57]. Our previous study also suggested that histidine metabolic disorders occurred in pregnant mice in the RA-induced CP model [51]. Therefore, whether histidine metabolism influences palate development and the precise role of ASVs within this dynamic warrant further exploration.

It can be seen that during palate development, ASVs and MSVs had basically similar regulatory effects on the palatal cells, but their specific regulatory pathways were different, which was due to their different sources and different miRNA components. The GO analysis showed that metabolic processes were among the top 10 biological processes enriched by miRNAs that were DE in the same stages between ASVs and MSVs, which suggested that ASVs and MSVs may have different effects on fetal cells by modulating metabolism. In line with this, we noted that the ASVs were involved in the metabolism regulation like histidine metabolism and retinoic acid metabolism during palate development. We suspected that this metabolism regulation might be mainly related to SEVs derived from fetal urine and lung fluid.

From a physiological standpoint, in this study, we examined normal embryos at various stages of palate development and revealed that ASV and MSV, particularly certain key miRNAs (miR-744-5p, miR-323-5p, and miR-3102-5p) within them, have the ability to regulate the functioning of palatal process cells. And we proposed that the aberrant alterations of these miRNAs could potentially serve as innovative predictive biomarkers for anomalies in palate development.

Furthermore, ASVs and MSVs may exhibit potential in modulating the functions of MEPM and MEE cells for the treatment of CP. Presently, SEVs, particularly mesenchymal stem cell (MSC) SEVs, have demonstrated promising therapeutic outcomes in tissue repair owing to their robust regenerative and immunomodulatory properties [58]. However, the application of SEV-based treatment for fetal abnormalities remains limited. Given the shared signaling pathways and gene regulatory networks between palate fusion and wound repair [59], it is proposed that the utilization of SEVs in interfering with palate development can be analogous to their application in tissue regeneration in future studies. Furthermore, SEVs can be modified to transport more cargo, such as the key miRNAs identified in our study, drugs, and enzymes, through techniques like electroporation, chemical-based transfection, and simple incubation methods, thereby enhancing their regenerative and immunomodulatory effects [60]. The highly targeted and efficient delivery of therapeutic factors could dramatically facilitate treatment efficiency. Despite there being many problems to explore, the use of SEVs as diagnostic and therapeutic tools for CP is promising and inspiring.

There are also some limitations to this study. First of all, since sample 1 in the E13.5 MSVs was collected from pregnant mice used to isolate MEPM cells, while other samples were collected uniformly before miRNA-Seq, sample 1 was seemingly different from the other two samples in the group due to a longer storage time and the degradation of some components. But even so, a large number of DE miRNAs were still identified via inter-group comparison. Second, since the differences in the miRNAs within the MSVs became limited after E14.5, few regulatory pathways have been identified. Samples over a longer time span during palate development should be collected to more fully understand the role of maternal SEVs in the early phase or late phase. Finally, although mice are the most conventional animal models to investigate the etiology of palate development and CP, there are some limitations, as their palate development cannot mimic the anatomical, morphological, and physiological features as thoroughly as humans, and the most obvious limitation is the developmental period [47]. It is known that human palatogenesis occurs in the early embryonic period, while in mice, this process occurs at the late–middle stage of mouse embryogenesis, so it will be better if the relationship between human body fluid SEVs and palate development can be explored, if available.

## 4. Materials and Methods

### 4.1. Animals

The pregnant mice at the E13.5/E14.5/E15.5 stages were sourced from Sibeifu Biotechnology Co., Ltd., Beijing, China. For each time point, 50 pregnant mice were collected. Among them, 24 pregnant mice, divided into 3 groups (8 mice in each group) according to the embryonic day, were used to isolate ASVs and MSVs for miRNA-Seq, and 26 pregnant mice were used to isolate ASVs and MSVs for in vitro cellular experiments (7–9 mice in each group). Each pregnant mouse had about 10–12 embryos. Two embryos from each pregnant mouse were selected for histological staining to observe whether the embryonic time was accurate (16 embryos from 8 pregnant mice at each time point), and the palate tissues of the remaining embryos in 50 pregnant mice were used for transcriptome-Seq and MEPM cell isolation. All animal experimentations were approved by the Animal Care and Use Committee at Beijing Stomatological Hospital, affiliated with Capital Medical University (permit number: KQYY-202208-003, Beijing, China).

### 4.2. Collection of AF and MP

AM and MP samples were obtained from pregnant ICR mice at the E13.5/E14.5/E15.5 stages. For MP collection, the mice were anesthetized using chloroform and then immediately dissected. Maternal blood from each group was collected from the inferior vena cava into tubes containing ethylenediaminetetraacetic acid. MP samples were separated via centrifugation at 1900× *g* for 10 min, and then the supernatant underwent further centrifugation at 3000× *g* for 10 min using a refrigerated centrifuge. For AF collection, the uterus was exposed, and AF was collected by inserting a 29.5-gauge needle directly into the amniotic cavity, taking care to prevent the puncture of the vitelline or allantoic vessels [19]. The AF and MP samples were either used immediately or partitioned into 1.5 mL microtubes and preserved at −80 °C.

### 4.3. Hematoxylin and Eosin (HE) Staining

To observe whether the embryonic time was accurate, two embryos from each pregnant mouse were selected, and the embryos’ heads were fixed with 4% paraformaldehyde for 48 h and dehydrated with gradient ethanol. Palate tissue histological sections were sliced to a thickness of 5 µm using a rotary microtome (Leica, Wetzlar, Germany) from the nasal cavity to the ear. The sections were stained with HE following methods described previously [22] and observed under an optical microscope.

### 4.4. SEV Isolation and Characterization

SEVs were isolated using differential centrifugation, as outlined in previous studies [19,61]. AF and MP samples underwent centrifugation at 300× *g* for 10 min and then 2000× *g* for 10 min to remove any exiting cells and dead cells. Then, the resulting supernatant was centrifuged at 10,000× *g* for 20 min to eliminate cellular debris. SEVs were subsequently separated using a Beckmann ultracentrifuge at 100,000× *g* for 4 h (MAX-XP, Beckmann, Palmer, AK, USA). The final pellet was resuspended in PBS. The SEVs were verified using transmission electron microscope (TEM), nanoparticle tracking analysis (NTA), and Western blotting. For each time point, ASVs and MSVs were collected from 24 pregnant mice and divided into 3 samples for miRNA analysis.

### 4.5. Isolation and Culture of MEPM Cells and MEE Cells

MEPM cells were isolated from pregnant ICR mice at E13.5 and cultured as previously reported [22]. The MEE cell line (BFN607200453, DSMZ Cell Bank, Shanghai, China) were obtained from the cell bank of the Chinese Academy of Sciences, and they were cultured in DMEM medium (SH30022.01, HyClone, Logan, UT, USA) and supplemented with 15% FBS (10099-141, Gibco, Miami, FL, USA) and 1% penicillin-streptomycin [17].

### 4.6. CCK8 Assay

Third-generation MEPM cells were plated into 96-well plates at a density of 7 × 10^3^ cells/well with six independent replicates. After the cells adhered to the wall, 10 μL CCK8 reagent (CK04, Dojindo Laboratories, Kumamoto, Japan) was added separately to each well at 0, 24, 48, and 72 h, followed by incubation at 37 °C for 2 h. Each sample was analyzed at 450 nm with a SpectraMax Paradigm microplate reader (10822-512, Molecular Devices, San Jose, CA, USA) [62].

### 4.7. Apoptosis Assay

Approximately 3 × 10^5^ cells/well were seeded in six-well plates and incubated at 37 °C. The following day, the cells were harvested using trypsin digestion without EDTA, and then stained with annexin V-fluorescein isothiocyanate (FITC) and PI following the instructions of the FITC-Annexin V Apoptosis Detection Kit (C1062L, Beyotime, Nantong, China). The cells were analyzed via flow cytometry (Accuri C6, BD, Washington, DC, USA) [62].

### 4.8. Scratch Test

Approximately 3 × 10^5^ cells/well were seeded into six-well plates. Scratch tests were performed as previously reported [22]. The cell migration distance was measured and calculated using Image-Pro Plus 6.0.

### 4.9. Alizarin Red Staining

Alizarin red staining was used to assess calcium mineralization in MEPM cells after 28 days of culture in an osteogenic induction medium. MEPM cells were fixed for 30 min and then stained with 1% alizarin red (A5533, Sigma-Aldrich, St. Louis, MO, USA) at room temperature. For quantification, calcium mineralization was measured as described in prior studies in the literature [62].

### 4.10. Western Blot

Protein samples were resolved on 10% SDS–PAGE gels and subsequently transferred onto polyvinylidene difluoride membranes. Membranes were blocked and then incubated with primary antibodies, including anti-CD9 (1:1000, 20597-1-AP, Proteintech, Rosemont, IL, USA), anti-CD63 (1:1000, 67605-1-1g, Proteintech, Rosemont, IL, USA), and anti-TSG101 (1:1000, 28283-1-AP, Proteintech, Rosemont, IL, USA), followed by incubating with appropriate horseradish peroxidase-conjugated goat-anti-rabbit antibody or goat-anti-mouse antibody. Proteins were detected via enhanced chemiluminescence detection reagents (P10300, NCM Biotech, Suzhou, China) [62].

### 4.11. Library Preparation and Sequencing

Total RNA containing miRNA was extracted and purified from enriched fractions of MSVs or ASVs using the miRNeasy^®^ Mini Kit (Qiagen, cat. no. 217,004) following the manufacturer’s protocol. For the small RNA libraries, 2.5 ng RNA from each sample was used to prepare RNA samples. Sequencing libraries were constructed using the NEB Next Multiplex Small RNA Library Prep Set for Illumina (NEB, Ipswich, MA, USA), following the manufacturer’s guidelines, and index codes were incorporated to assign sequences to their respective samples [9]. For miRNA, bands around ~150 bp were extracted. Then, the library’s quality was assessed using both the Agilent Bioanalyzer 2100 and qPCR. Clustering of the index-coded samples was carried out using a cBot Cluster Generation System with the TruSeq PE Cluster Kit v3-cBot-HS (Illumina, San Diego, CA, USA) as per the manufacturer’s instructions. After clustering, the libraries were sequenced on an Illumina Hiseq X Ten platform, generating 150 bp paired-end reads.

### 4.12. Quantification and DE Analysis of miRNAs

Clean reads were aligned using Bowtie2 software (2.4.4) and cross-referenced with database sequences from Silva, GtRNAdb, Rfam, and Repbase. Reads exhibiting > 10% N of suboptimal quality, with lengths > 32 nt/<16 nt, or having a 3′ adapter trimmed from their ends (no mismatch allowed), were excluded. After filtering out undesired sequences, the remaining reads were employed to identify known miRNAs and new miRNAs by comparing with miRbase (https://mirbase.org/, accessed on 12 January 2023). Randfold tools soft (http://rna.tbi.univie.ac.at/cgi-bin/RNAfold.cgi, accessed on 12 January 2023) was used for novel miRNA secondary structure prediction. DE miRNAs were identified based on a *p* value of ≤0.05 and a fold change (FC) of ≥2 using the edgeR R package (3.12.1). The raw data were uploaded to NCBI (accession number: PRJNA1027630).

### 4.13. Transcriptome-Seq Analysis

Total RNA was extracted from the fetal mouse palate (E13.5/E14.5/E15.5) using the RNeasy Micro Kit (QIAGEN, Hilden, Germany) according to the manufacturer’s instructions. The RNA samples were sequenced using the illumina NovaseqTM 6000 (LC Bio Technology CO., Ltd., Hangzhou, China). The mRNA expression levels were standardized using fragments per kilobase of exon model per million mapped reads. DE mRNAs were identified based on a *p* value of ≤0.05 and a fold change (FC) of ≥2. The raw data were uploaded to NCBI (accession number: PRJNA1027758).

### 4.14. GO and KEGG Enrichment Analysis

For functional enrichment analysis, all DE miRNA target genes were mapped to terms in the GO databases (http://www.geneontology.org/, accessed on 3 February 2023), and significantly enriched GO terms among the target genes were identified at a threshold of *p* < 0.05. GO term analyses were divided into three groups including cellular component, biological process, and molecular function. DE miRNAs were also mapped to the KEGG database (http://www.genome.jp/kegg/, accessed on 3 February 2023) to identify significantly enriched KEGG pathways at *p* < 0.05. Bar and bubble charts were constructed and visualized using the online platform at https://www.bioinformatics.com.cn (accessed on 3 February 2023).

### 4.15. Curation of miRNA in SEVs and mRNAs in Palate Tissue Regulation Pairs

MiRNA–gene regulation data were curated from several miRNA target databases including miRTarBase (http://mirtarbase.mbc.nctu.edu.tw/php/index.php, accessed on 17 February 2023), TargetScan (https://www.targetscan.org/vert_72/, accessed on 17 February 2023), and miRDB (http://mirdb.org/, accessed on 17 February 2023). Correlations between each miRNA and gene pair were computed. Only pairs with significant correlations were included in the network (correlation coefficient < 0, adjusted *p* value < 0.05). The network was constructed and visualized using Cytoscape v3.7.2 (https://cytoscape.org, accessed on 19 February 2023).

### 4.16. Validation of miRNA Expression

Expressions were analyzed using miRNA Universal SYBR qPCR Master Mix (MR101-01, Vazyme, Nanjing, China). The final results were normalized to U6 levels and analyzed by calculating the comparative cycle threshold values (2^−ΔΔCt^). All primer sequences are listed in Appendix A.

### 4.17. Statistical Analysis

All experiments were performed independently three times, and all statistical analyses were performed using Prism software (GraphPad Software 9.4.0). An unpaired two-tailed Student’s *t*-test was used to determine significance between two groups of normally distributed data. For comparisons between multiple groups, an ordinary one-way or two-way ANOVA was used, followed by Tukey’s test. The statistical tests used for each experiment are indicated in the figure legends. A value of *p* < 0.05 was considered statistically significant.

## 5. Conclusions

In conclusion, our investigation suggests that some pivotal biological activities such as cell proliferation, migration, osteogenesis, and apoptosis during palate development may be directed partly by stage-specific MSVs and ASVs. And the different stages of ASVs and MSVs, esp. some key miRNAs inside them such as miR-744-5p, miR-323-5p, and miR-3102-5p, both individually and synergistically, may possess diagnostic or therapeutic potential for diverse palate developmental anomalies.

## Figures and Tables

**Figure 1 ijms-24-17173-f001:**
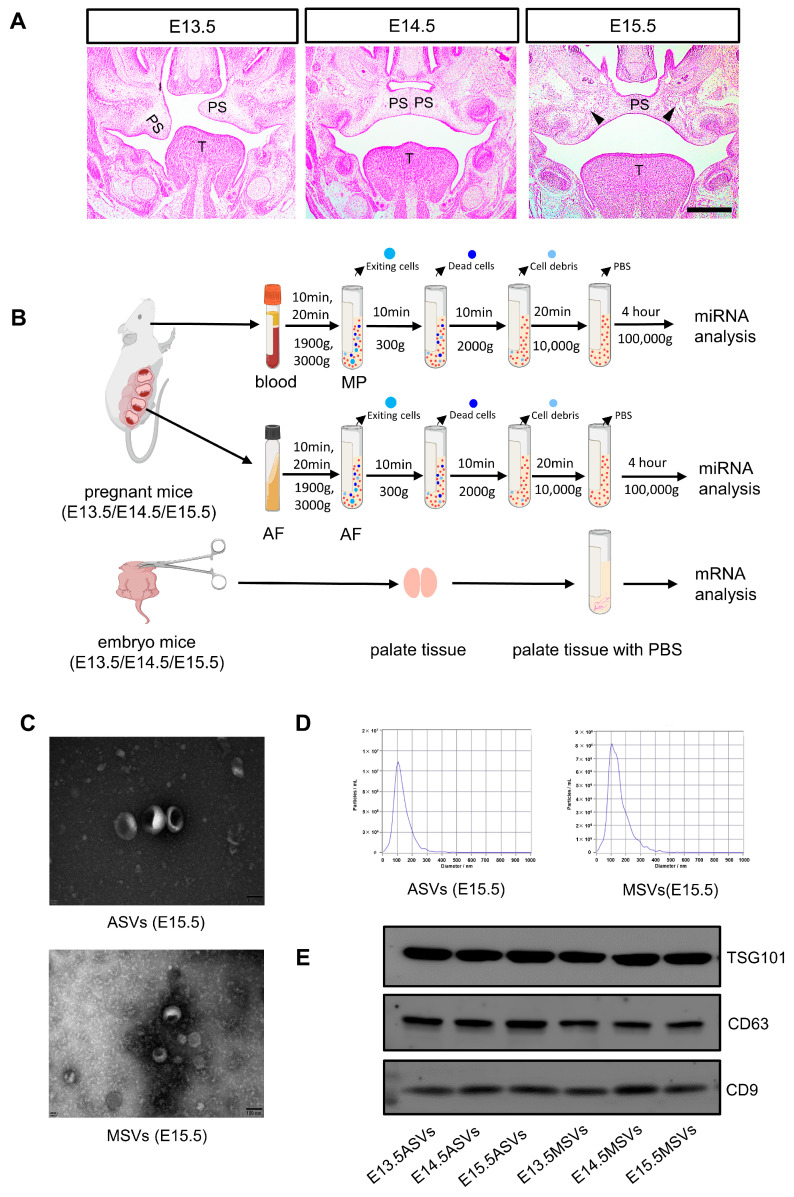
Isolation and characterization of ASVs and MSVs. (**A**) HE staining was used to observe the morphological changes during the development of palatal shelves in fetal mice at E13.5, E14.5, and E15.5. Scale bar, 200 μm. PS: Palatal shelves; T: tongue; the arrowheads indicate the osteogenetic center. (**B**) The workflow for isolating ASVs and MSVs via ultracentrifugation. (**C**) Transmission electron microscope (TEM) of ASVs and MSVs. Scale bar, 100 nm. (**D**) Nanoparticle tracking analysis (NTA) of ASVs and MSVs. (**E**) Western blot of ASV and MSV samples. CD63, CD9, and TSG101 were used as SEVs markers.

**Figure 2 ijms-24-17173-f002:**
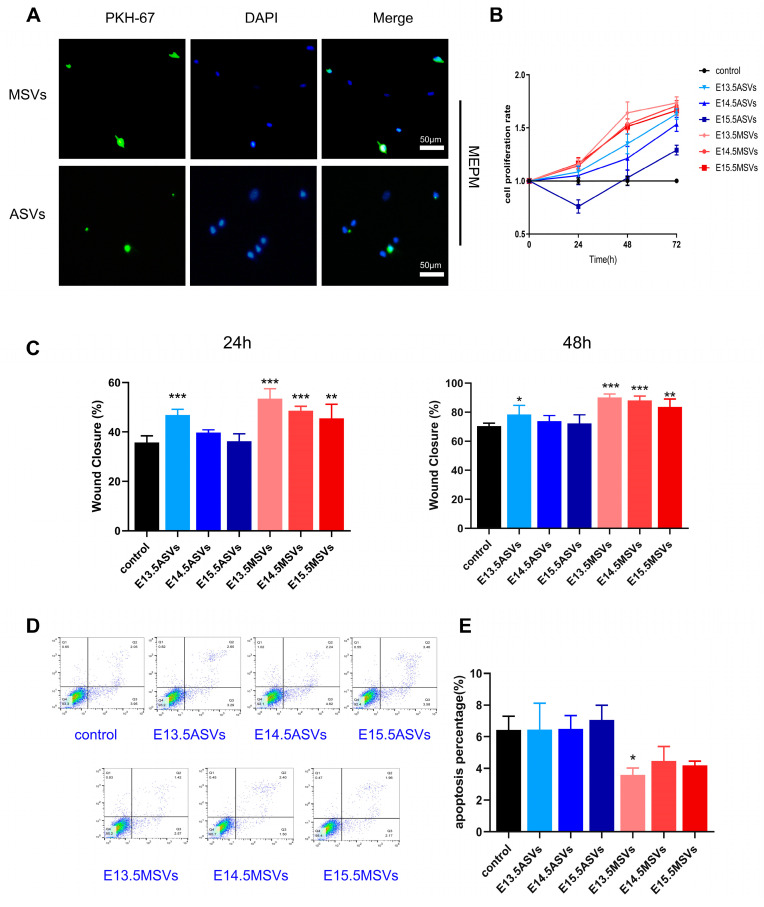
Effects of ASVs/MSVs from different palate developmental stages on proliferation, migration, and apoptosis of MEPM cells. (**A**) PKH67-labeled ASVs/MSVs could be uptaken by MEPM cells after being incubated for 24 h. Nuclei were counterstained with DAPI. Scale bars, 50 μm. (**B**–**E**) MEPM cells were treated with different groups of SEVs (100 μg/mL) for further study. (**B**) The proliferation rate determined using the CCK8 kit (*n* = 6). (**C**) Scratch assay for MEPM cell migration and cell migration rate were quantified by calculating the wound width (*n* = 9). (**D**,**E**) The early apoptotic rate determined via flow cytometry analysis with annexin V and PI staining (*n* = 3). The data are shown as the mean ± SD and were statistically analyzed via one-way ANOVA with Tukey’s multiple comparison test. All of the *p* values were two-sided, and adjustments were made for multiple comparisons. * *p* < 0.05, ** *p* < 0.01, *** *p* < 0.001.

**Figure 3 ijms-24-17173-f003:**
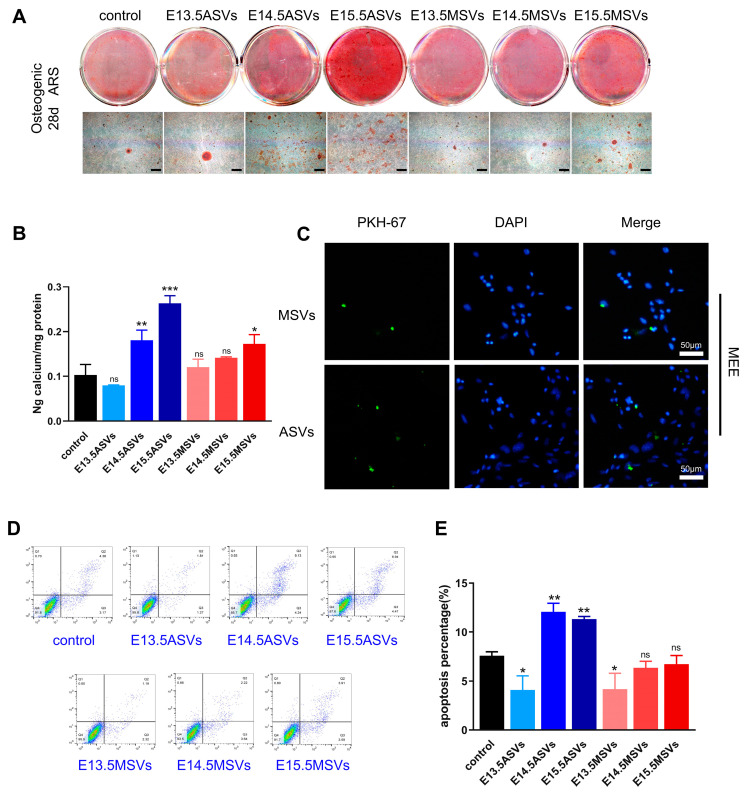
Effects of ASVs/MSVs from different palate developmental stages on osteogenesis of MEPM cells and apoptosis of MEE cells. (**A**) Alizarin red S staining on 28 days and (**B**) the calcium concentration was determined by measuring the absorbance at 562 nm on a multiplate reader (*n* = 3). Scale bars, 100 μm. (**C**) PKH67-labeled ASVs/MSVs could be uptaken by MEE cells after being incubated for 24 h. Nuclei were counterstained with DAPI. Scale bars, 50 μm. (**D**,**E**) MEE cells were treated with different groups of SEVs (100 μg/mL); AnnexinV-FITC/PI flow cytometric analysis and statistical data of MEE cell apoptosis after 48 h treatment (*n* = 3). The data are shown as the mean ± SD and were statistically analyzed via one-way ANOVA with Tukey’s multiple comparison test. All of the *p* values were two-sided, and adjustments were made for multiple comparisons. * *p* < 0.05, ** *p* < 0.01, *** *p* < 0.001. ns: not significant.

**Figure 4 ijms-24-17173-f004:**
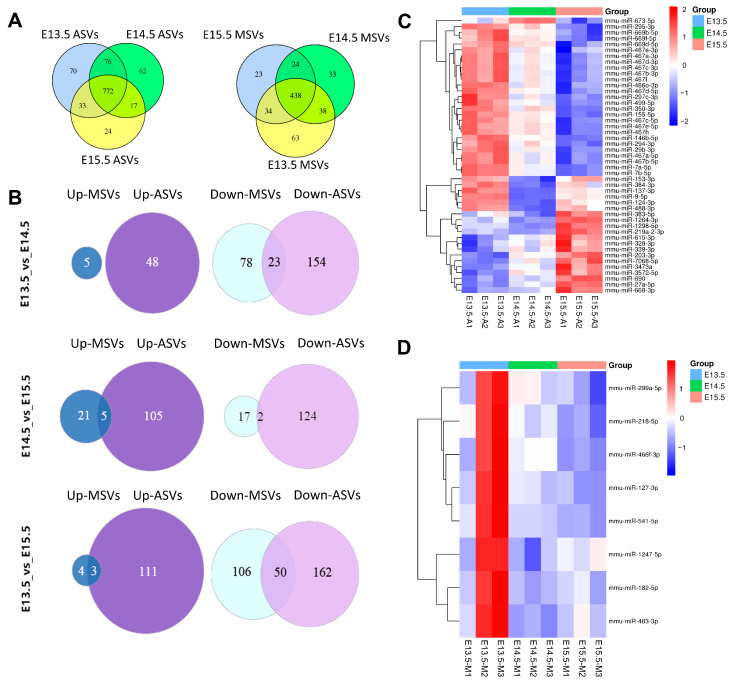
MiRNA expressing profiles in ASVs/MSVs from different palate developmental stages. (**A**) Venn diagrams showing the stage-specific miRNA expression profiling of ASVs or MSVs at E13.5/E14.5/E15.5, with each section showing the miRNA expression number. (**B**) Venn diagrams showing the numbers of DE miRNAs between E13.5/E14.5/E15.5 ASVs and MSVs via pairwise comparison. (**C**) Heatmap showing differential miRNA expression levels of ASVs from E13.5 to E15.5. (**D**) Heatmap showing differential miRNA expression levels of MSVs from E13.5 to E15.5.

**Figure 5 ijms-24-17173-f005:**
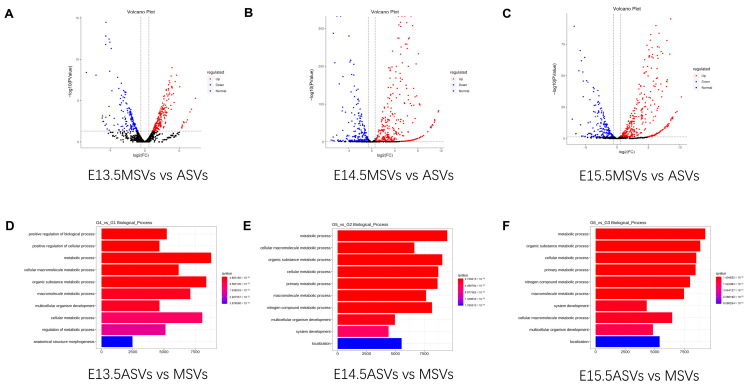
Analysis of DE miRNAs between ASVs and MSVs. (**A**–**C**) Volcano plots demonstrating the fold change (X-axis) and *p*-value (Y-axis) of the miRNAs between ASVs and MSVs at (**A**) E13.5, (**B**) E14.5, and (**C**) E15.5. Red and blue dots indicate the DE miRNAs, while the grey dots indicate the non-DE miRNAs. (**D**) Biological processes enriched in the DE miRNAs between E13.5ASVs vs. MSVs via GO assay. (**E**) Biological processes enriched in the DE miRNAs between E14.5ASVs vs. MSVs via GO assay. (**F**) Biological processes enriched in the DE miRNAs between E15.5ASVs vs. MSVs via GO assay.

**Figure 6 ijms-24-17173-f006:**
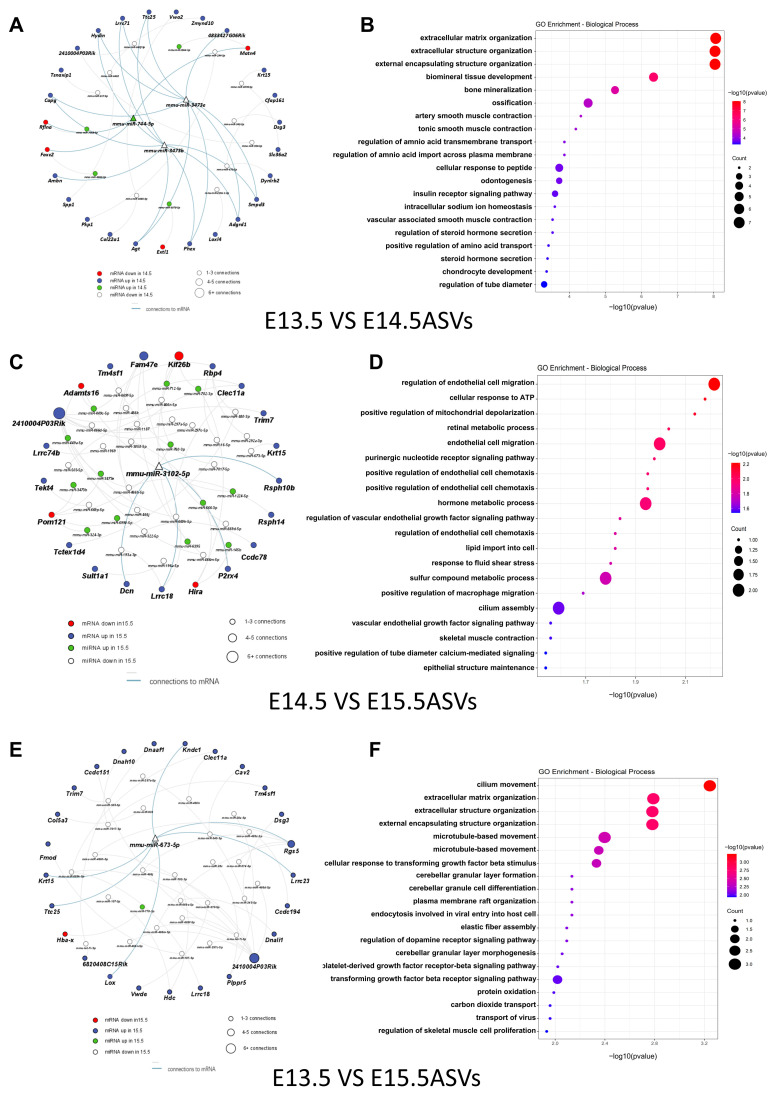
Regulatory networks of DE miRNA in ASVs and DE mRNAs in palate tissues. (**A**) Interaction network of DE miRNAs in ASVs and DE mRNAs in palate tissue that exhibited a negative correlation in “E13.5 vs. E14.5” comparison group. Size of node represents number of connections. Blue nodes represent upregulated mRNAs, red nodes represent downregulated mRNAs, green nodes represent upregulated miRNAs, and white nodes represent downregulated miRNAs. Triangles represent hub miRNAs. (**B**) Bubble plots depict the biological processes enriched in the (**A**) regulatory network via GO assay. Color scheme reflects the adjusted *p* values. The area of a circle is proportional to the number of enriched genes. (**C**) Interaction network of DE miRNAs in ASVs and DE mRNAs in palate tissue that exhibited a negative correlation in “E14.5 vs. E15.5” comparison group. (**D**) Bubble plots depict the biological processes enriched in the (**C**) regulatory network via GO assay. (**E**) Interaction network of DE miRNAs in ASVs and DE mRNAs in palate tissue that exhibited a negative correlation in “E13.5 vs. E15.5” comparison group. (**F**) Bubble plots depict the biological processes enriched in the (**E**) regulatory network via GO assay.

**Figure 7 ijms-24-17173-f007:**
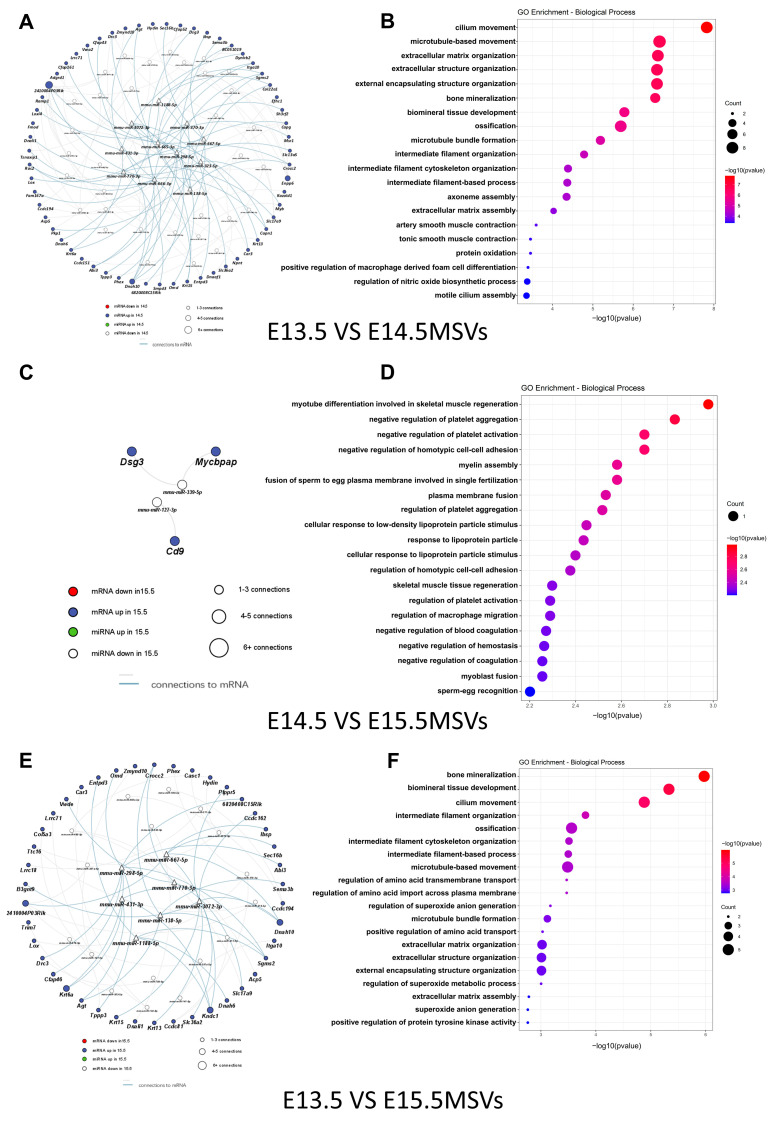
Regulatory networks of DE miRNA in MSVs and DE mRNAs in palate tissues. (**A**) Interaction network of DE miRNAs in MSVs and DE mRNAs in palate tissue that exhibited negative correlations in the “E13.5 vs. E14.5” comparison group. (**B**) Bubble plots depict the biological processes enriched in the (**A**) regulatory network via GO assay. (**C**) Interaction network of DE miRNAs in MSVs and DE mRNAs in palate tissue that exhibited a negative correlation in “E14.5 vs. E15.5” comparison group. (**D**) Bubble plots depict the biological processes enriched in the (**C**) regulatory network via GO assay. (**E**) Interaction network of DE miRNAs in MSVs and DE mRNAs in palate tissue that exhibited a negative correlation in “E13.5 vs. E15.5” comparison group. (**F**) Bubble plots depict the biological processes enriched in the (**E**) regulatory network via GO assay.

**Figure 8 ijms-24-17173-f008:**
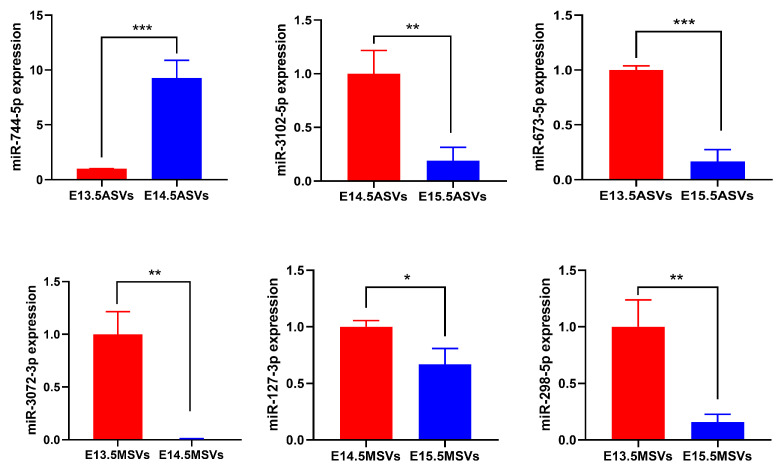
The verification of six DE miRNAs using qRT-PCR. The data are shown as the mean ± SD and were statistically analyzed via two-tailed Student’s *t*-test. * *p* < 0.05, ** *p* < 0.01, *** *p* < 0.001.

## Data Availability

All datasets generated for this study are included in the article/Appendix A; further inquiries can be directed to the corresponding author.

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
