# Peer review of "MicroRNAs in Small Extracellular Vesicles from Amniotic Fluid and Maternal Plasma Associated with Fetal Palate Development in Mice"

_ijms, 2023, doi:10.3390/ijms242417173_

Round 1

Reviewer 1 Report

Comments and Suggestions for Authors

The authors have submitted “MicroRNAs in small extracellular vesicles from amniotic fluid and maternal plasma associated with foetal palate development in mice”.

This is an interesting article with some point of interest for the scientific community; nevertheless, there are a few points to be improved before further considerations.

a.     Author have based their paper on the investigation on the role of MicroRNAs from amniotic fluid and maternal plasma in regenerative medicine. Nevertheless, they should increase the discussion, critically reporting the role of cell niche and the role of interactome in the clinical environment (Please, see and discuss: “Exosomes from Human Periapical Cyst-MSCs: Theranostic Application in Parkinson's Disease. International journal of medical sciences, 17(5), 657–663”.

b.     Also, in the discussion, authors should also give a brief discussion on the role of stem cells and their paracrine activity (Please, see and discuss: “Strategic Tools in Regenerative and Translational Dentistry”. (2019) International journal of molecular sciences. 2019; 20(8), 1879.), emphasizing the role of secretome even in the early stages of inflammatory conditions, comparing the immunomodulatory effects in different tissues/diseases.

c.       Please explain all the acronyms throughout the text.

d.       Some typos throughout the text should be corrected.

e.       In conclusion, the paper should highlight pros related to the potential applications on clinical tasks.

Author Response

Thank you very much for reviewing our manuscript “MicroRNAs in small extracellular vesicles from amniotic fluid and maternal plasma associated with foetal palate development in mice”. We are very grateful for your helpful feedback on our work. In response to your suggestions, we have revised the article and indicated these changes in the manuscript. In addition, the following are responses to your comments.

Reviewer 1:

The authors have submitted “MicroRNAs in small extracellular vesicles from amniotic fluid and maternal plasma associated with foetal palate development in mice”.

This is an interesting article with some point of interest for the scientific community; nevertheless, there are a few points to be improved before further considerations.

Point 1: Author have based their paper on the investigation on the role of MicroRNAs from amniotic fluid and maternal plasma in regenerative medicine. Nevertheless, they should increase the discussion, critically reporting the role of cell niche and the role of interactome in the clinical environment (Please, see and discuss: “Exosomes from Human Periapical Cyst-MSCs: Theranostic Application in Parkinson's Disease. International journal of medical sciences, 17(5), 657–663”.

Response 1: Thank you very much for your valuable suggestions. We have discussed them as follows labeled with red color in line 326-355: Recently, SEVs including their content were detected to communicate with cells by working as shuttles of molecules to influence target cells with biomarkers (miRNAs, proteins, lipids) which can be released in the extracellular environment, both in pathological and physiological conditions. Exploring specific SEV content which plays a key role in the pathophysiological conditions of patients or developing exosomes as a shuttle for the local drug-releasing and their content for targeted therapy is promising in the clinic[33]. In a study conducted by Jia et al., the analysis of MSVs from pregnant women carrying either normal fetuses or fetuses with CLP revealed the identification of let-7 clusters as prenatal biomarkers for CLP[16], which meanest the importance of SEVs of pregnant women in detecting embryo’s CLP.

In line 463-468: From a physiological standpoint, in this study, we further examined normal embryos at various stages of palate development and revealed that ASV and MSV, particularly certain key miRNAs (miR-744-5p, miR-323-5p, and miR-3102-5p) within them, have the ability to regulate the functioning of palatal process cells. And we propose that aberrant alterations of these miRNAs could potentially serve as innovative predictive biomarkers for anomalies in palate development.

Reference:

33.Tatullo, M.; Marrelli, B.; Zullo, M. J.; Codispoti, B.; Paduano, F.; Benincasa, C.; Fortunato, F.; Scacco, S.; Zavan, B.; Cocco, T., Exosomes from Human Periapical Cyst-MSCs: Theranostic Application in Parkinson's Disease. International journal of medical sciences 2020, 17, (5), 657-663.

16.Jia, S.; Zhang, Q.; Wang, Y.; Wei, X.; Gu, H.; Liu, D.; Ma, W.; He, Y.; Luo, W.; Yuan, Z., Identification by RNA-Seq of let-7 clusters as prenatal biomarkers for nonsyndromic cleft lip with palate. Annals of the New York Academy of Sciences 2022, 1516, (1), 234-246.

Point 2: Also, in the discussion, authors should also give a brief discussion on the role of stem cells and their paracrine activity (Please, see and discuss: “Strategic Tools in Regenerative and Translational Dentistry”. (2019) International journal of molecular sciences. 2019; 20(8), 1879.), emphasizing the role of secretome even in the early stages of inflammatory conditions, comparing the immunomodulatory effects in different tissues/diseases.

Response 2: Thank you sincerely for your valuable suggestions. We have discussed them in line 469-483 as follows labeled with red color: Furthermore, ASVs and MSVs may exhibit potential in modulating the function of MEPM and MEE cells for the treatment of CP. Presently, SEVs, particularly mesenchymal stem cells (MSCs)-SEVs, have demonstrated promising therapeutic outcomes in tissue repair owing to their robust regenerative and immunomodulatory properties[58]. However, the application of SEV-based treatment for foetal abnormalities remains limited. Given the shared signaling pathways and gene regulatory networks between palate fusion and wound repair[59], it is proposed that the utilization of SEVs in interfering with palate development can be analogous to their application in tissue regeneration in future studies. Furthermore, SEVs can be modified to transport more cargo, such as key miRNAs identified in our study, drugs, and enzymes, through techniques like electroporation, chemical-based transfection, and simple incubation methods, thereby enhancing their regenerative and immunomodulatory effects[60]. The highly targeted and efficient delivery of therapeutic factors could dramatically facilitate treatment efficiency. Despite many problems to explore, the use of SEVs as diagnostic and therapeutic tools for CP is promising and inspiring.

Reference:

58.Tatullo, M.; Codispoti, B.; Paduano, F.; Nuzzolese, M.; Makeeva, I., Strategic Tools in Regenerative and Translational Dentistry. International journal of molecular sciences 2019, 20, (8).

59.Biggs, L. C.; Goudy, S. L.; Dunnwald, M., Palatogenesis and cutaneous repair: A two-headed coin. Developmental dynamics : an official publication of the American Association of Anatomists 2015, 244, (3), 289-310.

60.Chen, M.; Xie, Y.; Luo, Y.; Xie, Y.; Wu, N.; Peng, S.; Chen, Q., Exosomes-a potential indicator and mediator of cleft lip and palate: a narrative review. Annals of translational medicine 2021, 9, (18), 1485.

Point 3: Please explain all the acronyms throughout the text.

Response 3: Thank you very much for your comments. We have added them in our revised manuscript and attached an abbreviation table at the end of our revised manuscript.

Point 4: Some typos throughout the text should be corrected.

Response 4: Thank you sincerely for your valuable suggestions. We have corrected them in our revised manuscript.

Point 5: In conclusion, the paper should highlight pros related to the potential applications on clinical tasks.

Response 5: Thank you very much for the valuable suggestions. We have revised the conclusion in our revised manuscript in line 639-644 labeled with red color: In conclusion, our investigation suggests that some pivotal biological activities such as cell proliferation, migration, osteogenesis and apoptosis during palate development may be directed partly by stage-specific MSVs and ASVs. And the different stages of ASVs and MSVs, esp. some key miRNAs inside them such as miR-744-5p, miR-323-5p and miR-3102-5p, both individually and synergistically, may possess diagnostic or therapeutic potential for diverse palate developmental anomalies.

Reviewer 2 Report

Comments and Suggestions for Authors

Dear Authors,

The mansucript deals with the SEV from amniotic fluid/maternal plasma in association to the foetal palate in mice. This is an experimanetal work marking new and original direction in the field. Thus, the manuscript contains new and relevant information. However, I have some remarks that should be improved mandatory for this manuscript:

1) I noticed that authors have demonstrated the routine histological staining haematoxylin and eosin for the slides in Figs. 1A. However, the method of obtaining this material (methodology of obtaining the material, fixation, preparing of slides, staining, evaluation) is not described in the manuscript. Please, include this info along the number of pregnant mice/obtained embryons used for the research! This is important!

2) in Materials and methods part subsection 4.1, 4.3 and commonly (24? its mentioned only in one place... and not in the beginning) - there is absence of info about number of animals used. Please, include this. If suddenly it was only limited number of mice, please, give provement about your data reliability... So, my questions - how many pregnant animals were used altogether? How many embryons for each stages in palate development were used? How the palate developmental stages were detected in embryons (mention please the methodology or inclusion/exclusion criteria). How many embryons for Fig. 1 was obtained commonly, and how many were valid? 

3) Please, try to give references for all the methods you have used (for instance, 4.3, 4.4., 4.5., 4.8. and in other aubsections as well).

4) Discussion. Please, remove references for the Fogs, as this is not a Result section and here uou have to discuss only your results. My main question here, - the interpretation of animal data always is slightly problematic for the human. Mice oral cavity differs from the developing human oral cavity... So, please, discuss this a little bit and indicate this as limitation of your manuscript at least.

5) Conclusions. Please, shorten and make them more precise. Remove first two extra useless sentences. Actually conclusions start from the 3rd sentence, - so, give short understanding about your findings in "different stages of ASVs and MSVs" and continue with your conclusion. (The last sentence would fit at the end of Discussion after limitations, where you can mention future perspectines, but not in conclusions).

References OK

Author Response

Thank you very much for reviewing our manuscript “MicroRNAs in small extracellular vesicles from amniotic fluid and maternal plasma associated with foetal palate development in mice”. We are very grateful for your helpful feedback on our work. In response to your suggestions, we have revised the article and indicated these changes in the manuscript. In addition, the following are responses to your comments.

Reviewer 2: Dear Authors,

The mansucript deals with the SEV from amniotic fluid/maternal plasma in association to the foetal palate in mice. This is an experimanetal work marking new and original direction in the field. Thus, the manuscript contains new and relevant information. However, I have some remarks that should be improved mandatory for this manuscript:

Point 1: I noticed that authors have demonstrated the routine histological staining haematoxylin and eosin for the slides in Figs. 1A. However, the method of obtaining this material (methodology of obtaining the material, fixation, preparing of slides, staining, evaluation) is not described in the manuscript. Please, include this info along the number of pregnant mice/obtained embryons used for the research! This is important!

Response 1: Thank you very much for your valuable suggestions. Actually, for each time point, 50 pregnant mice with 10-12 embryos each were collected as the amount of AM and MP samples in each pregnant mouse and their embryos was little. Among them, 24 pregnant mice divided into 3 groups (8 mice in each group) were used to isolate ASVs and MSVs for miRNA-Seq, and 26 pregnant mice were used to isolate ASVs and MSVs for in vitro cellular experiments (7-9 mice in each group). Two embryos from each pregnant mouse were selected for histological staining (haematoxylin and eosin, HE staining) to observe whether the embryonic time was accurate (16 embryos from 8 pregnant mice at each time point), and the palate tissues of the remaining embryos in 50 pregnant mice were used for transcriptome-Seq and MEPM cell isolation. For HE staining, the embryos’ heads were fixed with 4% paraformaldehyde for 48 h and dehydrated by gradient ethanol. Palate tissue histological coronal sections were sliced to a thickness of 5 µm using a rotary microtome from the nasal cavity to the ear. The sections were stained with HE following methods described previously [22] and observed under an optical microscope. We have added the method as “4.1. Animals and 4.3. Haematoxylin and eosin (HE) staining” labeled with red color in line 501-513 and 525-531 in our revised manuscript.

Reference

  1. Peng, X.; Chen, J.; Wang, Y.; Wang, X.; Zhao, X.; Zheng, X.; Wang, Z.; Yuan, D.; Du, J., Osteogenic microenvironment affects palatal development through glycolysis. Differentiation; research in biological diversity 2023, 133, 1-11

Point 2: in Materials and methods part subsection 4.1, 4.3 and commonly (24? its mentioned only in one place... and not in the beginning) - there is absence of info about number of animals used. Please, include this. If suddenly it was only limited number of mice, please, give provement about your data reliability... So, my questions - how many pregnant animals were used altogether? How many embryons for each stages in palate development were used? How the palate developmental stages were detected in embryons (mention please the methodology or inclusion/exclusion criteria). How many embryons for Fig. 1 was obtained commonly, and how many were valid?

Response 2: Thank you sincerely for your valuable suggestions. We have added them in “4.1. Animals” as follows labeled with red color in line 502-513. The pregnant mice at the E13.5/E14.5/E15.5 stages were sourced from Sibeifu Biotechnology Co., Ltd, Beijing, China. For each time point, 50 pregnant mice were collected. Among them, 24 pregnant mice divided into 3 groups (8 mice each group) according to the embryonic day were used to isolate ASVs and MSVs for miRNA-Seq, and 26 pregnant mice were used to isolate ASVs and MSVs for in vitro cellular experiments (7-9 mice each group). Each pregnant mouse had about 10-12 embryos. Two embryos from each pregnant mouse were selected for histological staining to observe whether the embryonic time was accurate (16 embryos from 8 pregnant mice at each time point), and the palate tissues of the remaining embryos in 50 pregnant mice were used for transcriptome-Seq and MEPM cell isolation. All animal experimentations were approved by the Animal Care and Use Committee at Beijing Stomatological Hospital, affiliated with Capital Medical University (permit number: KQYY-202208-003, Beijing, China).

Point 3: Please, try to give references for all the methods you have used (for instance, 4.3, 4.4., 4.5., 4.8. and in other aubsections as well).

Response 3: Thank you very much for the valuable suggestions. We have added them in our revised manuscript.

Point 4: Discussion. Please, remove references for the Fogs, as this is not a Result section and here uou have to discuss only your results. My main question here, - the interpretation of animal data always is slightly problematic for the human. Mice oral cavity differs from the developing human oral cavity... So, please, discuss this a little bit and indicate this as limitation of your manuscript at least.

Response 4: Thank you very much for the valuable suggestions. We have added them in line 492-499 as follows labeled with red color: Finally, although mice are the most conventional animal models to investigate the etiology of palate development and CP, there are some limitations as their palate development can’t mimic the anatomical, morphological, and physiological features thoroughly as humans, and the most obvious limitation is the developmental period[47]. It is known that human palatogenesis occurs in the early embryonic period, while in mice, this process occurs at the late-middle stage of mouse embryogenesis, so it will be better that the relationship between human body fluid SEVs and palate development can be explored if available.

Reference

  1. Liu, J.; Chen, J.; Yuan, D.; Sun, L.; Fan, Z.; Wang, S.; Du, J., Dynamic mRNA Expression Analysis of the Secondary Palatal Morphogenesis in Miniature Pigs. International journal of molecular sciences 2019, 20, (17).

Point 5: Conclusions. Please, shorten and make them more precise. Remove first two extra useless sentences. Actually conclusions start from the 3rd sentence, - so, give short understanding about your findings in "different stages of ASVs and MSVs" and continue with your conclusion. (The last sentence would fit at the end of Discussion after limitations, where you can mention future perspectines, but not in conclusions).

Response 5: Thank you very much for the valuable suggestions. We have removed the first two sentences and moved the last sentence to the end of the discussion. The revised conclusion in line 639-644 as follows labeled with red color: In conclusion, our investigation suggests that some pivotal biological activities such as cell proliferation, migration, osteogenesis and apoptosis during palate development may be directed partly by stage-specific MSVs and ASVs. And the different stages of ASVs and MSVs, esp. some key miRNAs inside them such as miR-744-5p, miR-323-5p and miR-3102-5p, both individually and synergistically, may possess diagnostic or therapeutic potential for diverse palate developmental anomalies.

Round 2

Reviewer 1 Report

Comments and Suggestions for Authors

no further comments